# Menstrual Cycle-Related Hormonal Fluctuations in ADHD: Effect on Cognitive Functioning—A Narrative Review

**DOI:** 10.3390/jcm15010121

**Published:** 2025-12-24

**Authors:** Dora Wynchank, Regina M. G. T. M. F. Sutrisno, Emma van Andel, J. J. Sandra Kooij

**Affiliations:** 1Expertise Center Adult ADHD, PsyQ, Parnassia Group, Carel Reinierszkade 197, 2593 HR The Hague, The Netherlands; e.vanandel@psyq.nl (E.v.A.); s.kooij@psyq.nl (J.J.S.K.); 2Department of Cognitive and Systems Neuroscience, Faculty of Science, University of Amsterdam, Sciencepark 904, 1098 XH Amsterdam, The Netherlands; 3Amsterdam University Medical Center, Department of Psychiatry, Vrije Universiteit Amsterdam, De Boelelaan 1117, 1081 HV Amsterdam, The Netherlands

**Keywords:** ADHD, menstrual cycle, ovarian hormones, oestrogen, progesterone, oestradiol, premenstrual syndrome, premenstrual dysphoric disorder, cognitive function, attention, executive function, inhibitory control, impulsivity

## Abstract

Attention-Deficit/Hyperactivity Disorder (ADHD) is a common neurodevelopmental disorder linked to impaired cognition and altered dopamine neurotransmission. Emerging evidence suggests that women with ADHD experience pronounced hormone-related difficulties, with menstrual cycle-related changes in mood and cognition interfering with daily functioning and diminishing treatment efficacy. This review examines the influence of hormonal fluctuations during the menstrual cycle on cognitive functioning and ADHD symptomatology in women. A comprehensive literature search of Ovid EmBase identified studies published between 2015 and 2025 examining cognitive performance, including attention, executive functioning, working memory, and inhibitory control, across menstrual cycle phases in women with or without ADHD. Twenty-nine studies met inclusion criteria. Neurobiological measurements included hormonal assays, neuroimaging, and neurotransmitter models. Seven studies in non-clinical populations suggested that attentional processing was enhanced during the mid-luteal phase, which may be linked to higher progesterone levels. By contrast, four studies in women with ADHD and six studies in women with mood-related disorders, such as PMS or PMDD, consistently observed impairments in attention, executive function, and impulsivity during the mid-luteal and pre-menstrual phases. These objective findings parallel subjective reports of worsened cognition, heightened mood symptoms, and diminished medication efficacy during the luteal phase. Current evidence indicates that ADHD-related cognitive functioning fluctuates with the menstrual cycle, with impairments particularly evident in women with ADHD and/or comorbid mood disorders. These changes may reflect increased sensitivity to allopregnanolone, peri-menstrual oestrogen withdrawal, and the absence of compensatory neural adaptations observed in non-clinical populations. However, findings remain preliminary and sometimes contradictory due to methodological heterogeneity and small sample sizes. Further research is needed to clarify these mechanisms and, importantly, to translate theoretical insights into clinical application through female-specific diagnostic procedures and treatment strategies.

## 1. Introduction

Attention-deficit Hyperactivity Disorder (ADHD) is a common neurodevelopmental and heritable disorder characterised by persistent patterns of childhood onset inattention and/or hyperactivity-impulsivity that hinder daily functioning across the lifespan. Girls and women (hereafter referred to as ‘women’, defined as those assigned female at birth and menstruating) are diagnosed with ADHD less often and on average four years later than boys/men [1,2,3,4]. Women are also more likely to present inattention, internalising symptoms, and comorbidities such as anxiety or depression, whereas men more often show hyperactivity, externalising behaviours, and related comorbidities [1,2,5].

ADHD affects several cognitive domains, such as executive function, attention, working memory and inhibitory control [6,7,8], with some evidence of sex differences [9]. Executive functions regulate goal-directed behaviour by coordinating lower-level processes including attention, working memory, inhibition and task-switching [10]. Attentional processes can be either stimulus-driven (i.e., bottom-up) or guided by prior knowledge (i.e., top-down) [11] and are further distinguished into several subtypes; selective, divided, and sustained attention [12]. Working memory entails the cognitive ability to temporarily hold and manipulate new information in the short term, and is necessary for complex tasks like reasoning, learning, and problem-solving [13]. ADHD symptoms and deficits in cognitive functioning are associated with altered dopamine neurotransmission in the brain—particularly in the mesocortical, mesolimbic and nigrostriatal pathways, which are involved in cognitive and attentional processes [14,15,16]. As these dopaminergic systems are sensitive to ovarian-hormone fluctuations, they provide a plausible route through which hormonal variation may modulate cognitive performance in women with ADHD.

Dopaminergic signalling is also affected by cyclical fluctuations in oestrogen and progesterone, which rise and fall across the follicular, ovulatory, and luteal phases [17,18] (See Figure 1). While these ovarian hormones influence dopamine activity, cognitive functioning, and mood [19,20,21,22], their effects in women with ADHD remain poorly understood. Emerging data suggest they may experience more severe hormone-related mood and cognitive symptoms than women without ADHD [9,23], with debilitating consequences for daily life [24]. This interaction between hormonal signalling and dopaminergic pathways provides a neurobiological basis for cycle-related variation in cognitive performance. Two mechanistic pathways have been proposed to explain these fluctuations: an oestrogen-withdrawal model affecting dopaminergic signalling, and an allopregnanolone-sensitivity model involving altered GABAergic modulation. This review compares the extent to which current evidence supports each pathway. A clearer understanding of these fluctuations is crucial for developing personalised treatment approaches. The specific knowledge gap this review addresses is the absence of an integrated synthesis linking menstrual-cycle hormonal variation with domain-specific cognitive changes and ADHD symptom variability, while also assessing the different methodologies that shape these findings. To our knowledge, no previous review has systematically synthesised evidence on how cyclical hormonal fluctuations across the menstrual cycle influence cognitive functioning in individuals with ADHD. The aim of this review is to integrate findings across cognitive domains and menstrual phases, identify patterns and methodological heterogeneity in the existing literature, and highlight implications for clinical management and future research. This narrative review synthesises peer-reviewed studies published between 2000 and 2025 that examined cognitive performance or related neuropsychological outcomes across menstrual-cycle phases, in individuals with and without ADHD. Electronic searches were mainly conducted in Ovid EmBase using combinations of terms relating to ADHD, menstrual cycle, and cognition. Both clinical and experimental studies were included if they reported cognitive or attentional outcomes in relation to endogenous hormonal fluctuations. The review focuses on human data, excluding animal studies and articles without cycle-phase specification, to delineate the current boundaries of inference. Accordingly, this review examines how menstrual cycle–related hormonal fluctuations influence cognitive functioning and ADHD symptom severity in women.

## 2. Methods

This literature review examined how the menstrual cycle affects cognitive symptoms related to ADHD in women. Studies comparing the influence of fluctuating reproductive hormones on ADHD symptoms in those with and without ADHD were investigated. A comprehensive search for relevant studies was conducted in Ovid EmBase from February 2025, using the following search string: (ADHD OR attention deficit hyperactivity disorder OR ADHD symptoms OR attention OR impulsivity OR hyperactivity OR executive function OR working memory OR cognitive inhibition) AND (hormones OR reproductive hormones OR sex hormones OR progesterone OR *estrogen OR *estradiol OR testosterone OR menstruation OR menstrual cycle OR menstrual period). In addition, reference lists of the studies included were also examined for potential articles. The search was restricted to articles based on human research published in English between 2015 and 2025.

Inclusion criteria for the studies were: empirical studies of women during the menstrual cycle, both those with and without an ADHD diagnosis, provided they either had ADHD symptoms measured by valid ADHD questionnaires or were examined for cognitive functions related to ADHD. ADHD-related cognitive functions were defined as (in)attention, executive functioning, working memory, inhibition and/or impulsivity. Hormones were defined as oestrogen, progesterone, and testosterone. Menstrual cycle phases were determined based on the phases defined in Schmalenberger and colleagues (2021) [18]—which distinguishes between the pre-menstrual (day 24 to 28), menstrual (day 1 to 7), mid-follicular (day 4 to 8), pre-ovulatory (day 12 to 13), post-ovulatory (day 15–17), and mid-luteal (day 20 to 24) phases. Phase-verification methods varied across studies and included self-reported cycle tracking, ovulation tests, and salivary or blood hormone assays; this heterogeneity was taken into account when interpreting findings.

Duplicates were removed automatically in Ovid and manually cross-checked before screening. Screening of titles and abstracts was conducted by one reviewer and verified by a second reviewer. Full-text eligibility decisions and data extraction were performed by the first reviewer and subsequently reviewed by the second reviewer to ensure accuracy. Results were presented as a narrative review of the relevant literature according to PRISMA-ScR guidelines [25]. Study quality was assessed using structured methodological indicators rather than a formal risk-of-bias tool, because of the heterogeneity of designs included. Each study was evaluated for sample-size adequacy, clarity of cognitive-task operationalisation, hormone-verification method (self-report, ovulation testing, or salivary/blood assays), and appropriateness of study design. These indicators served as pragmatic markers of methodological robustness and informed the interpretation of heterogeneous findings.

## 3. Results

Of 343 studies screened, 70 were assessed for eligibility and 29 met the inclusion criteria (see Table 1; for a summary of findings per menstrual cycle, see Appendix A). Excluded studies either failed to meet these criteria or used different definitions for menstrual cycle phases and/or did not distinguish between different hormonal events (Figure 2) [18].

### 3.1. Attention

Most evidence from non-clinical samples suggested that attentional processes were reported as faster and more accurate during the mid-luteal phase compared to the other menstrual phases, reflected in increased alertness [26], attention allocation processes [27], faster reaction times [28,29,30] and increased accuracy [31,32] during the mid-luteal phase. Greater distractibility towards incongruent [26] or social stimuli [29,31] was present in the mid-luteal phase, compared with the menstrual, mid-follicular or pre-ovulatory phases [26,29,32]. This heightened attentional sensitivity was associated with elevated progesterone in the mid-luteal phase [26,28,31]. However, findings were mixed: one study associated higher progesterone with slower reaction times [31], while another found no correlation [27].

Findings on divided attention in non-clinical samples were mixed as well: one study reported faster responses and fewer errors in a dual-task paradigm during the mid-luteal phase [30], whereas another found slower and less accurate performance compared to women in the early follicular phase or men [33]. Moreover, sustained attention seemed to be impaired during periods of elevated oestrogen levels, as higher error-rates were found on sustained attention tasks during the pre-ovulatory phase in one study [34] and mid-luteal phase in another [33].

Women with premenstrual syndrome (PMS) showed reduced attentional control on an attentional network test in the mid-luteal phase compared to the follicular phase [35]. In women with premenstrual dysphoric disorder (PMDD), attention was particularly impaired during the pre-menstrual phase relative to controls [36,37,38]. Subjectively, women with PMDD reported more inattention during the pre-ovulatory, mid-luteal, and pre-menstrual phase [37], and this was most correlated with PMDD impairment [36]. Objective deficits were mainly seen pre-menstrually [38]. 

Among the thirteen studies that assessed attentional outcomes in people without ADHD:Seven were non-clinical investigations which reported improved or faster attentional performance during mid-luteal phases;Two studies that examined divided attention showed mixed results;Two studies reported impaired sustained attention during elevated oestrogen periods;Four studies examining clinical samples, including PMS and PMDD, identified attentional decline or increased variability during the mid-luteal or pre-menstrual phase.

### 3.2. Executive Function

Most studies in non-clinical samples reported executive function changes during the mid-luteal phase, linked to progesterone levels [39,40]. Faster reaction times were found on a neutral word Stroop-task [39], while a face-gender Stroop-task showed slower but more accurate responses to female faces [40]. Neuroimaging findings further showed that the nodal efficiency of the left inferior frontal gyrus within the resting-state executive control network correlated with progesterone, indicating greater importance for network communication during the mid-luteal phase [40]. Lower error rates on a task-switching paradigm were also found in the mid-luteal phase, although performance did not interact with any cycle phases [32]. One study found improved overall cognitive scores during the menstrual phase, even though participants perceived that a poorer mood was affecting their cognitive performance [34]. Executive function also varied in women with PMS, who show greater emotional interference on a Stroop task pre-menstrually, but the opposite pattern during the mid-follicular phase [41]. In PMDD, executive function was impaired pre-menstrually, with poorer Simon’s task performance and less use of cognitive reappraisal for emotion regulation during both the mid-luteal and pre-menstrual phases [36].

Across the six studies assessing executive function outcomes:Four reported enhanced or more stable performance during the mid-luteal phase in non-clinical samples;Two studies identified performance decline during the pre-menstrual phase in women with PMS or PMDD.

#### Working Memory

Findings on working memory in non-clinical populations were mixed. Several studies reported no significant cycle-related differences [42,43,44]. One exception was Gaizauskaite and colleagues (2025) [42], who initially observed no significant differences among women in the mid-luteal phase, mid-follicular phase, or those using oral contraceptives or intrauterine devices. However, they later found that women in the mid-follicular phase performed worse than oral-contraceptive users on the n-back task, only when the task became more difficult. By contrast, Louis and colleagues (2023) [45] reported that within-person increases in oestrogen were linked to faster reaction times on the n-back test, regardless of task difficulty. One randomised crossover clinical trial reported a perimenstrual decline in working memory in participants with suicidal ideation, which was prevented with transdermal oestrogen administration [46]. When differences in working memory emerged, they often related to target detection: performance on the n-back test was enhanced in the mid-luteal phase [39,47,48], faster responses correlated with higher progesterone [39] and higher oestrogen levels [45], and greater hit rates and sensitivity [47,48]

Regarding menstrual cycle-dependent changes in brain connectivity, Hidalgo-Lopez and colleagues (2021) [43] found no changes in working memory across phases in a non-clinical sample, but observed that fronto-striatal connectivity with salience and effort-related regions varied: connectivity was low during menstruation, negative during pre-ovulatory phase and strongly positive during the mid-luteal phase, supporting greater interconnectivity and cue sensitivity. They also reported stronger suppression of visual areas in lure verses target trials of the n-back test and increased connectivity between dorsolateral prefrontal cortex (dlPFC) and posteromedial regions during the mid-luteal phase (while reduced in the pre-ovulatory phase).

Across nine studies that examined working memory outcomes:Four studies found no significant changes across menstrual cycle phases;Two studies observed contrasting effects of task-difficulty on working memory performance across the menstrual cycle;Three studies examining working memory found significant changes related to target detection during the mid-luteal phase;One study reported a perimenstrual decline in women with suicidal ideation, which was prevented with transdermal oestrogen administration.

### 3.3. Inhibition and Impulsivity

Findings on inhibition in non-clinical populations were mixed: several studies reported no cycle-related differences on behavioural inhibition tasks [34,43,46]. In contrast, Yen and colleagues (2023) [38] found impaired response inhibition on a go/no-go task in women with PMDD during both the post-ovulatory and pre-menstrual phases, which correlated with greater self-reported impulsivity and irritability only in the pre-menstrual phase.

Impulsivity findings were also mixed. Some studies reported increases around ovulation (objectively during the pre-ovulatory phase [54], subjectively during the post-ovulatory phase [38,53]), while others reported increases around menstruation (subjectively during pre-menstrual phase [37,38], objectively during the menstrual phase [51]). One of these studies further linked oestrogen levels during menstruation to more impulsive choices and poorer delay of reward on a reward acquisition paradigm, particularly in non-clinical women with low self-reported trait impulsivity [51]. In PMDD, self-reported impulsivity appears heightened post-ovulation and pre-menstrually [38], although one study noted that the pre-menstrual effect was clearer when women with ADHD were excluded [37]. One neuroimaging study in a non-clinical sample showed that right dlPFC activity was elevated relative to the dorsal striatum in the mid-luteal phase and was sensitive to oestrogen levels during a delayed discounting task [54]. Both regions form a key circuit for the cognitive control of impulsivity.

Out of 4 studies that examined inhibition across the menstrual cycle, 3 studies in non-clinical samples reported no cycle-related differences, and only 1 study found an impaired response inhibition during the post-ovulatory and pre-menstrual phase in women with PMDD.All 5 studies in non-clinical women and women with PMDD that assessed objective or subjective impulsivity found that it increased around either ovulation or menstruation.One neuroimaging study in a non-clinical sample showed that right dlPFC activity was elevated during the mid-luteal phase and was sensitive to oestrogen levels.

### 3.4. ADHD Symptoms

The evidence regarding ADHD symptoms across the menstrual cycle was limited and included few controlled studies, but nevertheless appeared to suggest that low oestrogen levels (during the mid-follicular, post-ovulatory and pre-menstrual phase) were linked to exacerbated symptoms, especially inattention [37,49,50,53]. In a non-clinical sample, Roberts and colleagues (2018) [53] found that this effect was strongest when progesterone levels were high, particularly in women with high self-reported trait impulsivity. Lin and colleagues (2024) [37] further showed that women with both PMDD and ADHD experienced more memory problems and dysfunctional impulsivity in the pre-ovulatory and mid-luteal phases compared to women with PMDD alone. 

Qualitative studies showed that women with ADHD reported worsening of ADHD and depressive symptoms in the mid-luteal [49] and (pre-)menstrual phase [49,50]. Reported problems include executive dysfunction, emotional dysregulation, and attentional difficulties. Moreover, women described that their ADHD medication became less effective around the mid-luteal and pre-menstrual phase [49,50]. A recent case series by De Jong and colleagues (2023) [50] administered individually tailored stimulant dosage increases in 9 female ADHD patients during the premenstrual week. Interestingly, all participants reported an improvement in ADHD symptoms, mood, and energy levels, which all matched their non-premenstrual baseline with little to no side effects.

One study in a non-clinical sample found that ADHD symptoms exacerbated when oestrogen levels were low and progesterone levels were high, especially in those with high-trait impulsivity.Another study observed that women with both PMDD and ADHD reported more memory problems and dysfunctional impulsivity during the pre-ovulatory and mid-luteal phases compared to women with PMDD alone.Two qualitative studies reported that women with ADHD experienced worsening ADHD and depressive symptoms during the mid-luteal and (pre-)menstrual phase, and a decrease in stimulant medication effectiveness.

## 4. Discussion

This literature review examines how ovarian hormone fluctuations across the menstrual cycle influence ADHD-related cognitive function and symptom severity. Overall, the evidence suggests that cognition and ADHD symptoms fluctuate across different phases of the menstrual cycle. This effect appears to be particularly pronounced in women with psychiatric conditions, who may be more susceptible to the influence of hormonal fluctuations on cognition [37,55]. In women with ADHD, exacerbation of symptoms is most consistently reported during the luteal phase (mid-luteal and pre-menstrual), marked by increased experiences of inattention and executive dysfunction [49,50]. These difficulties are especially evident in individuals with higher baseline impulsivity [53] and may be accompanied by reduced effectiveness of ADHD medication [49,50]. Similar impairments in attention, executive function, and emotional regulation are observed in women with mood-related disorders, such as PMS [35,41] or PMDD [36,37,38]. In contrast, studies in non-clinical populations mainly report improved executive functioning and working memory in the mid-luteal phase, possibly reflecting heightened sensitivity in attentional processes. Around ovulation, findings are less consistent with some evidence for impaired sustained attention and increased impulsive decision-making. Results during menstruation are also mixed, ranging from increased impulsivity [51] to improved overall cognitive scores [34].

Despite these converging mechanistic findings, interpretation of the literature is constrained by marked methodological heterogeneity. Sample sizes were often small, limiting statistical power and increasing the risk of false-positive or false-negative associations. Studies also varied widely in how menstrual-cycle phases were defined and verified; some relied on retrospective self-report or calendar counting, whereas others used salivary or blood hormone assays or ovulation testing. These differences are noted in Table 1 and summarised in its footnotes. Variation in cognitive-task selection, timing of assessment, and whether a single cycle or multiple cycles were examined further complicates comparison across studies. Together, these factors likely contribute to the mixed results observed in several cognitive domains and caution against overinterpreting phase-specific effects unless findings are replicated with rigorous hormonal verification and standardised cognitive protocols. 

Cognitive functioning may be impacted by ovarian hormones such as oestrogen and progesterone through their interactions with neurotransmitter signalling. Oestrogen shows agonist-like effects on dopaminergic signalling by enhancing its neurotransmission and modulating its receptor expression [17,56,57,58,59], especially in the prefrontal cortex [59,60] and striatum [17]—regions critical for cognitive function and decision-making [17,59,61]. Consequently, enhanced dopaminergic activity during menstrual phases with higher oestrogen (i.e., pre-ovulatory and mid-luteal [58]) may support improved cognitive function, whereas reduced activity during lower oestrogen phases [19,62], may contribute to impairment. On the other hand, progesterone interacts with multiple neurotransmitters: it facilitates inhibitory GABAergic transmission through the GABA receptors (particularly via its neuroactive metabolite, allopregnanolone), suppresses excitatory glutamate responses, and increases serotonin transmission by regulating its related gene and protein expression [17]. However, the effects of oestrogen and progesterone on cognition are complex and may depend on individual variations in baseline dopamine function [39,45] and receptor sensitivity [58]. The interaction between oestrogen and dopamine is also suggested to follow an inverted U-shape curve, in which optimal signalling and striatal function occur within a specific range [17,45].

### 4.1. Cognitive Function Across the Menstrual Cycle in ADHD

Current hypotheses suggest that ADHD stems from a hypofunctioning dopamine system due to increased reuptake of dopamine, resulting in reduced levels of extracellular dopamine [16]. Commonly used pharmacological treatments for ADHD aim to counteract this deficit by increasing brain catecholamine levels and inhibiting the reuptake of dopamine, thereby regulating dopaminergic signalling [14]. Interestingly, women seem to have a more sensitive dopamine system and a higher striatal dopamine receptor availability than men [63,64]. This may suggest that women with ADHD have an even higher reuptake and lower availability of dopamine than their male counterparts. Rapidly declining oestrogen levels during the pre-menstrual phase, combined with altered dopamine functioning, may then contribute to impairments in attention, executive function and inhibitory processes during the luteal phase.

Empirical studies support this pattern: ADHD symptom severity during the luteal phase has been linked to low oestrogen and high progesterone, especially in women with high baseline impulsivity [53]. Across studies, inattention and impulsivity are consistently elevated in the mid-luteal and pre-menstrual phases [37,49,50]. Qualitative work further describes intensified depressive symptoms, attentional difficulties, and reduced medication efficacy, all of which can be improved when stimulant dosages are adjusted pre-menstrually [49,50]. Taken together, these results support the hypothesis that ovarian hormone-related dopaminergic dysregulation is particularly severe during the luteal phase in women with ADHD, resulting in worsening symptoms and reduced medication efficacy.

Several of the empirical findings reviewed here map directly onto the two main mechanistic hypotheses proposed for menstrual cycle–related cognitive changes in ADHD. Firstly, behavioural evidence consistent with an oestrogen-withdrawal/dopamine-reduction mechanism includes luteal-phase declines in attention and inhibitory control observed in ADHD, PMS, and PMDD samples [35,36,37,38,41,49]. Although these studies do not directly measure dopaminergic activity, the late-luteal impairment pattern is compatible with models in which falling oestradiol reduces fronto-striatal dopamine availability, increasing cognitive vulnerability. These findings also align with neuroimaging evidence demonstrating that oestradiol enhances dopaminergic signalling and fronto-striatal engagement during cognitive-control tasks [12,33,43]. 

Secondly, findings from PMS/PMDD studies showing pronounced late-luteal affective and cognitive dysregulation [41,49] are consistent with the allopregnanolone-sensitivity hypothesis, whereby heightened responsiveness to progesterone-derived neurosteroids alters GABA-A receptor modulation. Neurobiological studies demonstrating altered GABAergic inhibition and differential neural responses to allopregnanolone challenge [65,66] provide stronger mechanistic support for this model. Notably, studies reporting cycle phase-dependent variation in fronto-striatal and salience-network activation offer the most direct mechanistic bridge between hormonal fluctuations and cognitive vulnerability in ADHD [33,43]. Together, these findings suggest that luteal-phase cognitive decline in ADHD may result from the combined effects of reduced dopaminergic tone as oestradiol falls and increased sensitivity to progesterone-derived neurosteroids in susceptible individuals. 

### 4.2. Cognitive Function Across the Menstrual Cycle in Relation to Mood

Similarly to women with ADHD, women who experience mood disorders such as PMS, PMDD or suicidal ideation appear to have more consistent and severe fluctuations across the menstrual cycle than women without these disorders—with attention and executive functioning worsening in the mid-luteal and pre-menstrual phases [35,36,37,38]. Interestingly, recent studies show that ADHD and PMDD show high comorbidity—approximately 45.5% of women with ADHD also experience PMDD symptoms [23], and ADHD prevalence is also significantly higher in women with PMDD [37]. This overlap highlights shared vulnerabilities, including inattention, impaired executive control, and heightened impulsivity. While inattention appears to be most strongly associated with overall PMDD severity [36], deficits in executive functioning appear more closely linked to emotion regulation. Women with PMS present different processing of emotional information [41] and women with PMDD show less cognitive reappraisal for emotion regulation during the luteal phase [36].

PMDD is thought to be related to a heightened sensitivity to allopregnanolone—a neurometabolite derived from progesterone [67,68]. Normally, allopregnanolone facilitates the inhibitory function of GABA [17]. On the other hand, in PMDD, normal levels of allopregnanolone seem to trigger negative mood symptoms related to hypersensitivity, such as irritability, stress sensitivity, anxiety, and social rejection sensitivity [62,67,68]. Luteal-bound impairments in attention and executive functioning in women with PMS [35] or PMDD [36,37,38] may reflect this hypersensitivity to allopregnanolone, as heightened distractibility, negative mood, and overstimulation could hinder cognitive functioning. Evidently, both impulsivity and irritability are heightened in PMDD during the pre-menstrual phase, and are linked to an impaired response inhibition [38]. This effect may be further reinforced, as weakened cognitive reappraisal during the mid-luteal phase impairs emotion regulation [36].

In addition to this luteal-bound pattern, studies in women with menstrual cycle-related mood disorders also suggest a sensitivity to rapidly declining oestrogen levels around menstruation [62]. As oestrogen facilitates dopaminergic signalling—and with it cognitive function [17,59,61,62]—lower oestrogen levels during the pre-menstrual phase may contribute to impairments in attention, executive function, and inhibitory processes, a pattern observed in women with PMDD [36,38]. The literature also suggests that the prefrontal cortex—a key brain region for cognitive functioning—is particularly sensitive to oestrogen fluctuations across the menstrual cycle [60] and disruptions in oestrogen signalling have been found in several psychiatric disorders [55,69]. Moreover, this may explain why administering oestrogen during the pre-menstrual phase prevents the decline in working memory performance observed in this phase [46].

Preliminary findings further suggest that comorbid ADHD and PMDD intensify these difficulties: women with both conditions report more daily memory problems and impulsivity than those with PMDD alone [37]. The combined effects of heightened allopregnanolone sensitivity, oestrogen withdrawal, and increased attentional sensitivity may amplify irritability, distractibility, and impairments in attention, executive functioning, and emotion regulation in women with both ADHD and PMDD.

Given the high comorbidity between ADHD and premenstrual disorders such as PMS and PMDD (e.g., Refs. [36,37]), future research should incorporate systematic assessment of premenstrual symptomatology when studying cognitive or emotional changes across the menstrual cycle in women with ADHD. Standardised screening instruments, such as the Daily Record of Severity of Problems (DRSP) for PMDD [70], or validated PMS rating scales, should be used to differentiate baseline ADHD-related symptoms from cycle-related mood and cognitive fluctuations. Incorporating these measures would allow studies to stratify participants by PMS/PMDD status or statistically control for premenstrual symptom severity, thereby reducing diagnostic confounding and strengthening the internal validity of menstrual-cycle research in ADHD. These recommendations are also consistent with emerging clinical guidance on managing female-specific ADHD presentations [71].

### 4.3. Cognitive Function Across the Menstrual Cycle in the General Population

In non-clinical populations, some studies report no significant menstrual cycle effects on cognitive functions impaired in ADHD [34,42,43,44,46], which is consistent with earlier literature suggesting that cycle-related fluctuations in cognitive performance are subtle or absent [72,73]. However, among studies that do find changes, the mid-luteal phase consistently stands out as the phase where cognitive differences are most apparent. During this phase, attentional processing and related executive functions are generally faster [26,27,28,30,39,47,48], more accurate [31,32], yet more susceptible to distraction [26,29,31], suggesting heightened alertness or information processing. Some studies link these changes to higher progesterone levels [26,28,31], which enhance inhibitory GABA-transmission in non-clinical populations, promoting information processing by suppressing irrelevant neural activity. However, other evidence reveals impairments in sustained attention during the mid-luteal phase [33], likely due to increased distractibility or oestrogen-related rises in impulsivity. Similar mid-luteal changes are seen in executive functioning and working memory, where both processing speed [39,47,48] and accuracy [32,40] are enhanced. These effects may be driven by improved attentional processing, since executive function relies on lower-level processes such as attention [10]. Supporting this idea, some studies argue that better executive task performance may reflect enhanced attentional sensitivity and processing speed, rather than improved executive functioning [32,34]. Together, these findings indicate that in non-clinical populations, attentional processing is particularly sensitive to the effects of the menstrual cycle.

Neuroimaging studies also suggest that information processing improves during the mid-luteal phase in non-clinical samples. Increased connectivity between frontostriatal regions and networks involved in salience and cognitive effort appears to boost attentional sensitivity [43]. The inferior frontal gyrus plays a central role in communication between brain areas when progesterone levels are high, which may further support efficient information processing [40]. Resting-state fMRI findings also show increased activity in the dlPFC during the mid-luteal phase, which may enhance cognitive control of impulsivity and contribute to better information processing. However, this activity seems to be more sensitive to oestrogen than to progesterone [54]. These changes in brain connectivity patterns are thought to act as a compensatory mechanism to maintain stable cognitive functioning throughout the menstrual cycle [43]. This may explain why menstrual cycle-related fluctuations in cognitive performance are subtle or absent in some studies. Improved cognitive control of impulsivity during the mid-luteal phase [54] may contribute to more efficient information processing and less distractibility by irrelevant stimuli, despite an increased attentional sensitivity.

However, this interpretation warrants caution. Several studies report cycle-related changes in fronto-striatal or salience-network connectivity without measurable behavioural differences, indicating that compensatory recruitment may not always preserve performance [43]. A more nuanced possibility is that such compensation is effective only under low-to-moderate cognitive load, but may fail under higher task demands or in individuals with underlying vulnerabilities, such as PMS/PMDD or ADHD, where baseline neural efficiency is reduced. Future research could test these hypotheses by varying task difficulty or stratifying participants by comorbidity to identify the conditions under which compensatory mechanisms succeed or fail. 

Findings around the ovulation period in non-clinical populations are less consistent. Some studies report impaired sustained attention [34] and more impulsive decision-making [54] during the pre-ovulatory phase. Neuroimaging data suggest that inhibitory control may also be diminished, since the dlPFC appears to exert more neural inhibition on the inferior frontal gyrus, which may reduce communication between brain areas and make information processing less efficient [43]. Evidence for cognitive changes during menstruation is even more inconsistent, with one study suggesting impairments in decision-making linked to increased oestrogen levels [51], in contrast to the more consistent luteal-phase effects.

These findings from non-clinical populations reveal that cognitive functioning related to ADHD symptoms appears to be affected by menstrual cycle-related hormonal fluctuations, although they are more subtle and inconsistent than in clinical (ADHD) populations. In non-clinical populations, such deficits may be stabilised by compensatory brain connectivity patterns that help uphold cognitive performance during the mid-luteal phase [43]. By contrast, women with ADHD may lack such compensatory mechanisms, making them more vulnerable to cycle-related hormonal changes and experiencing more pronounced deficits in attention, executive function, and medication effectiveness during this phase. Neuroimaging studies show that adults with ADHD recruit different brain areas for the same cognitive task, with reduced activation in task-relevant regions such as the inferior frontal cortex and altered connectivity in the striatum, cingulate, and cerebellum during motor response inhibition and working memory. At the same time, they show increased activation in default mode network regions, suggesting poor deactivation of this network contributes to distractibility [6,74,75]. This may explain why women with ADHD are more vulnerable to cycle-related hormonal changes, experiencing more pronounced deficits in attention and executive function and a reduced effectiveness of their medication during the luteal phase of the menstrual cycle.

### 4.4. Limitations and Future Considerations

Most studies are limited by small sample sizes and methodological inconsistencies, making it difficult to draw solid conclusions. First, many studies differ in their specific definition of the menstrual cycle phases they examine, or examine phases too generally without accounting for changing hormone levels within that phase. Future studies are therefore recommended to examine menstrual cycle effects based on the different hormonal events that occur within each phase and avoid using general phases such as the luteal or follicular phase.

Second, most studies are also inconsistent in their definition of cognitive functions. Studies either use certain parts of cognitive tests as indicators for specific cognitive functions, or the same cognitive test is used by different studies to measure different cognitive functions. For example, the Stroop task generally measures executive function, but has been used by different studies to reflect attentional or inhibitory processes instead. Even though these processes are part of executive functioning, it becomes difficult to compare findings on specific aspects of cognitive function if the same measure is used to represent different cognitive functions. Some also rely on subjective reports rather than objective tests, which may partly explain discrepancies in findings [34].

Third, some studies are unclear about how they accounted for learning effects. These are important to take into consideration, since repeated measurements over different menstrual cycle phases may cause learning effects that could explain improvements in performance. One study found that differences in cognitive function during the first cycle changed when they were examined again during a second cycle [44]. Future studies should therefore make sure to account for learning effects in their methodology by either randomising the first measurement or using multiple versions of the same test.

Fourth, findings may also be inconsistent due to individual variations in baseline hormone levels or neurotransmitter availability. The menstrual cycle is fundamentally a within-person process and should be treated as such in analyses [18]. The mid-follicular phase may be particularly vulnerable to individual variations in hormone levels. Here, there may be an earlier rise of pre-ovulatory oestrogen or low, stable oestrogen levels depending on the length of the menstrual cycle [18]. Some studies also find different hormonal effects depending on baseline dopamine availability [39,45] or baseline impulsivity [37,51,53]. Studies examining the menstrual cycle are therefore recommended to implement a within-subject design, take individual differences in dopamine levels into account, and examine multiple cycles instead of only one to ensure the effects are consistent rather than attributable to random variation.

Fifth, another limitation is that many neuroimaging studies show connectivity changes without behavioural differences, making compensatory interpretations uncertain.

Sixth, most studies did not assess or control for PMS or PMDD, despite their high comorbidity with ADHD. This limits the ability to distinguish ADHD-related cognitive patterns from cycle-related mood or attentional changes. Future studies should therefore screen for PMS/PMDD using validated tools such as the DRSP and adjust analyses accordingly.

In addition, this review has two methodological limitations related to the search and screening procedures. Only Ovid EmBase was used, so relevant studies indexed exclusively in other databases may have been missed. Also, a single primary reviewer conducted screening and data extraction. Although all decisions were verified by a second reviewer, this approach may still introduce some risk of selection or extraction bias. Cycle-informed medication adjustments should also be approached cautiously. Evidence remains limited to small case series and clinician observations, and there are no controlled trials to support routine use. Any dose modifications should therefore be piloted in carefully monitored clinical settings with systematic tracking of benefits and adverse effects. 

Lastly, the current literature review mainly examined cognitive functioning across the menstrual cycle and did not consider how emotion regulation and/or reward processing were affected, even though these are also known to be impacted by the menstrual cycle [19,62,76,77]. Type of information (social or emotional) and general stress levels also appear to impact cognitive functioning differently across the menstrual cycle [27,29,31,32,40,76,78,79], as well as PMDD symptoms [78]. Besides further investigating how hormones and the menstrual cycle affect women with ADHD, future studies should therefore also take the impact of menstrual cycle-related hormonal fluctuations on emotion regulation, social cognition or reward processing into account.

Beyond these methodological considerations, future research would benefit from incorporating validated biological markers and interdisciplinary approaches that can directly test the mechanistic hypotheses outlined in this review. Very few existing studies combine precise hormonal assays with neuroimaging or neurophysiological measures, limiting our ability to map fluctuations in oestradiol, progesterone, and neurosteroids onto dynamic changes in fronto-striatal and salience-network function. Integrating endocrinology, multimodal neuroimaging, and computational modelling would allow researchers to quantify hormone–brain–behaviour interactions more precisely, identify individual profiles of hormone sensitivity, and develop mechanistically grounded, personalised models of cycle-related cognitive vulnerability in ADHD.

### 4.5. Clinical Implications

Although research on ADHD in women is steadily increasing, it remains limited. The findings in this literature review support the need for a greater clinical understanding of menstrual cycle effects on ADHD symptoms and treatment response. Greater awareness may improve treatment outcomes through a better understanding of how fluctuations in ovarian hormones across the menstrual cycle can affect ADHD symptoms and medication efficacy. Women with ADHD frequently report worsening symptoms and reduced medication effectiveness, during the mid-luteal and pre-menstrual phases. Clinicians should therefore ask explicitly about these experiences during assessment and treatment follow-up, as this may reveal patterns that otherwise go unnoticed. Encouraging the use of cycle tracking, through diaries or digital applications, can help patients and providers identify when additional support is needed and facilitate personalised treatment strategies that address cycle-related changes in cognition, emotion regulation, and stimulant medication effectiveness.

From a treatment perspective, several strategies may be considered. Pre-menstrual adjustment of stimulant dosage, supported by recent preliminary case-series evidence, appears safe and can restore symptom control with minimal side effects [50]. Flexible addition of short-acting formulations may also be helpful for managing cyclic symptom peaks. Beyond pharmacological strategies, psychoeducation and patient support are equally important. Informing women that these fluctuations are hormonally driven can reduce self-blame, improve treatment adherence, and empower them to anticipate periods of greater vulnerability. A female-specific treatment group that addressed pre-menstrual worsening of ADHD and mood symptoms was evaluated qualitatively and found to be valuable: women reported feeling recognised, validated, and better able to prioritise self-acceptance and self-care [80]. Such programmes may represent a promising addition to standard ADHD treatment. Several trials are now underway to evaluate this strategy systematically. Only early exploratory feasibility work exists at present, and no formal clinical trials are underway.

Attention should also be given to comorbidities such as PMS and PMDD, which are common in women with ADHD [23] and may exacerbate cognitive and emotional difficulties. Early identification of these conditions allows for integrated treatment approaches, including lifestyle strategies, stress reduction, and—in some cases—hormonal or adjunctive interventions. Finally, clinicians and researchers should remain mindful that timing of assessments may influence diagnostic accuracy, as symptom severity and cognitive test performance can vary substantially across the cycle.

Ethical considerations are also important when applying menstrual cycle-informed ADHD care. Any medication adjustments should be used cautiously, within a shared decision-making framework, and with close monitoring for benefit and adverse effects. Clinicians should avoid adoption of unvalidated dosing strategies and remain mindful of gender-based diagnostic biases, ensuring that hormonal explanations do not undermine patient autonomy. These safeguards help ensure that emerging cycle-informed approaches are implemented responsibly and ethically. 

Together, these implications highlight that integrating menstrual cycle considerations into ADHD care can improve recognition, treatment tailoring, and ultimately quality of life for women.

## 5. Conclusions

This review shows that ovarian hormone fluctuations can influence cognitive functioning across the menstrual cycle, with effects most pronounced in women with ADHD and those with PMS or PMDD. While cognitive changes in non-clinical samples tend to be subtle or inconsistent, clinical populations consistently show luteal-phase worsening in attention, executive function, impulsivity, and medication effectiveness. These findings highlight the importance of incorporating menstrual-cycle awareness into assessment and treatment planning for women with ADHD. 

Although early case-series suggest that some women may benefit from cycle-informed medication adjustments, stronger empirical evidence is needed before such approaches can be recommended routinely.

Future research should prioritise:Within-subject, multi-cycle prospective designs;Standardised cognitive and hormonal measurement methods;Carefully monitored trials evaluating tailored medication strategies.

Strengthening methodological rigour in these areas will help establish safe, effective and individualised ADHD care for women across the menstrual cycle.

## Figures and Tables

**Figure 1 jcm-15-00121-f001:**
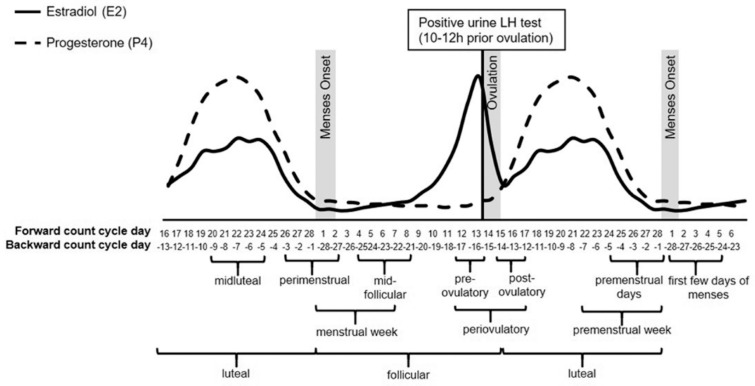
Graphical overview of the menstrual cycle and its phases, with the characteristic fluctuations of the ovarian hormones oestradiol (E2) and progesterone (P4)*. * Reprinted from Psychoneuroendocrinology, Vol/123, Schmalenberger et al. (2021) [18], How to study the menstrual cycle: Practical tools and recommendations, 104,895, Copyright (2025), with permission from Elsevier.

**Figure 2 jcm-15-00121-f002:**
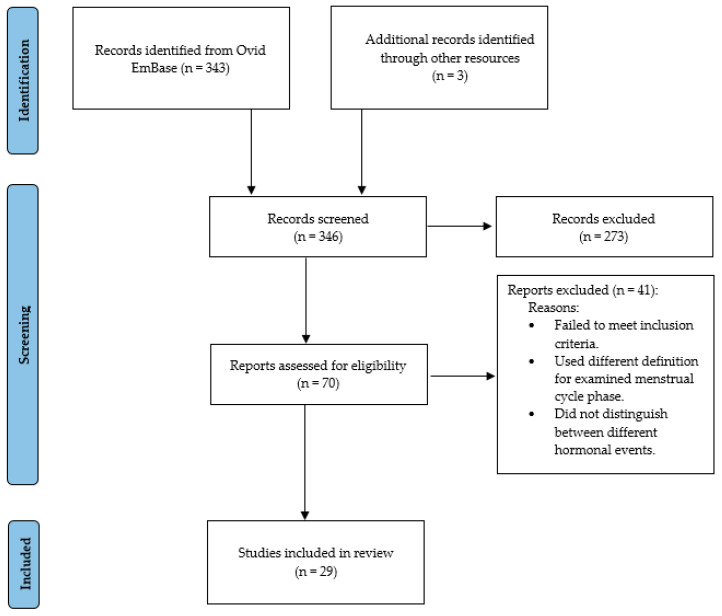
Flow diagram of study inclusion process.

**Table 1 jcm-15-00121-t001:** Summary of findings of included studies that examined the effect of the menstrual cycle on cognitive symptoms in ADHD.

Author (Year)	Subjects	Measured Domains (Instrument)	Menstrual Cycle Phases	Phase Verification Method	Main Results
Cohen et al. (2022) [26]	NCF = 21 OCF = 24	Attention (Attentional network test—interactions with alerting and no-alerting condition)	Mid-follicular Mid-luteal	Saliva samples	No significant differences between phases in attention in OCF group Mid-luteal phase: ↑Alertness mediated by P4 in NCF group, ↑Interference in incongruent trials
Pilarczyk et al. (2019) [27]	NCF = 20	Attention allocation to different stimuli (eye-tracking)	Mid-follicular Mid-luteal	Saliva samples Ovulation test	Mid-luteal phase: Faster RT, attention was allocated earlier to key regions of presented stimuli P4 levels did not correlate significantly with any measure of visual attention.
Brotzner et al. (2015) [28]	NCF = 18	Spatial attention (Visuospatial cued attention task)	Mid-follicular Pre-ovulatory Mid-luteal	Saliva samples Ovulation test	Mid-luteal phase: ↑Attention, RT correlated positively to P4 levels
Li and Deng (2022) [29]	NCF = 36	Social cognition (visual search task with social and object distractors) Attention (eye-tracking)	Pre-ovulatory Mid-luteal	Backwards counting ^a^	Pre-ovulatory phase: Slower RT, ↓Fixation on social distractors Mid-luteal phase: Faster RT, ↑Fixation social distractors
Alkanat et al. (2021) [30]	NCF = 40	Divided attention (Annett’s peg moving task + Go/no-go task)	Pre-ovulatory Mid-luteal	Ovulation test	Mid-luteal phase: ↑Divided attention
Wang and Chen (2020) [31]	NCF = 26	Attention (Attention network test) Emotional information processing (Emotional face flanker task)	Menstrual Pre-ovulatory Mid-luteal Pre-menstrual	Self-report tracking ^b^ Saliva samples	Mid-luteal phase: Slower RT, ↑Accuracy RT to sad faces correlated positively with P4 levels
Xu et al. (2022) [32]	NCF = 79	Cognitive flexibility (Task-switching paradigm) Divided attention (Audiovisual cross-modal monitoring task) Inhibition (spatial Stroop task) Working memory (Multiple change detection paradigm)	Menstrual Pre-ovulatory Mid-luteal	Backwards counting ^a^	Mid-luteal: ↑Sensitivity on divided attention task, ↑Cognitive flexibility
Pletzer et al. (2017) [33]	Males = 35 NCF = 32	Selective, divided and sustained attention (d2-R, FAIR-2, sustained attention test from the Wiener Test system)	Mid-follicular Mid-luteal	Backwards counting ^b^ Ovulation test	Mid-luteal phase: ↓Selective/divided attention, ↓Sustained attention in NCF
Ronca et al. (2025) [34]	Males = 96 NCF = 105 OCF = 47	Sustained attention, inhibition (Smiley task battery) Reaction time (Spatial simple reaction time) Visuospatial function (Cube Analysis test) Mood (Burgess Brief Mood Questionnaire)	Menstrual Mid-follicular Pre-ovulatory Luteal	Self-report tracking ^a^ Backwards counting ^a^	Menstrual phase: ↑Cognitive performance, faster RT, ↓Mood in NCF Pre-ovulatory phase: ↓Sustained attention in NCF
Blaser et al. (2024) [35]	Low PMS = 36 High PMS = 29	Attention (Attentional network test-R with emotional stimuli)	Mid-follicular Mid-luteal Pre-menstrual	Forward and backwards counting ^a^	Mid-luteal phase: ↓Attentional control in high PMS
Lin et al. (2022) [36]	PMDD = 63 NCF = 53	Executive function (Simon Task) Attention (Attention and Performance Self-Assessment scale) Fatigue (Fatigue Severity Scale) Insomnia (Pittsburgh Insomnia Rating Scale) Depression (The Center for Epidemiological Studies’ Depression Scale) Emotion regulation (Emotion Regulation Questionnaire)	Mid-luteal Pre-menstrual	Self-report tracking ^a^	Mid-luteal: ↓Cognitive reappraisal of emotions in PMDD Pre-menstrual: ↓Executive function, ↓Attention, ↓Cognitive reappraisal of emotions, ↑Insomnia, ↑Fatigue in PMDD Inattention was the most associated factor of PMDD functional impairment
Lin et al. (2024) [37]	PMDD = 58 No PMDD = 50	ADHD symptoms (psychiatric assessment) Attention (Attention and Performance Self-Assessment scale) Impulsivity (Dickman Impulsivity Inventory)	Pre-menstrual Pre-ovulatory Mid-luteal	Self-report tracking ^b^ Ovulation test	Diagnostic criteria for ADHD were met significantly more often in PMDD group Pre-menstrual phase: ↓Attention, ↑Impulsivity in PMDD only Pre-ovulatory phase: ↓Attention, ↑Impulsivity in PMDD (with ADHD) Mid-luteal phase: ↓Attention
Yen et al. (2023) [38]	PMDD = 63 No PMDD = 53	Inhibition (Go/no-go task) Attention (Go trials in Go/no-go task) Impulsivity (Dickman’s Impulsivity Inventory)	Post-ovulatory Pre-menstrual	Self-report tracking ^a^	Post-ovulatory phase: ↓Inhibition, ↑Impulsivity in PMDD Pre-menstrual phase: ↓Inhibition, ↓Attention, ↑Impulsivity in PMDD
Hidalgo-Lopez and Pletzer (2017) [39]	NCF = 36	WM (n-back test) Executive function (Stroop task)	Menstrual Pre-ovulatory Mid-luteal	Saliva samples	Mid-luteal phase: ↑WM when ↑P4, faster RT on Stroop-task when ↑P4, ↑Baseline DA was related to ↓Inhibition, ↓Eyeblink-rate was related to ↑Inhibition
Wang et al. (2025) [40]	Pre-ovulatory = 28 Mid-luteal = 25	Executive function (face-gender Stroop task) Resting-state fMRI Task-based fMRI	Pre-menstrual Mid-luteal	Self-report tracking ^b^ Saliva samples	Mid-luteal phase: ↑Accuracy for female face stimuli only P4 was positively correlated to differences in RT to female and male faces, and the nodal efficiency of inferior frontal gyrus in the resting-state executive control network
Eggert et al. (2017) [41]	PMS = 55 Non-PMS = 55	Executive Function (Emotional Stroop task)	Mid-follicular Pre-menstrual	Self-report tracking ^a^	Mid-follicular phase: ↑Emotional Stroop effect in non-PMS women Pre-menstrual phase: ↑Emotional Stroop effect in PMS women
Gaizauskaite et al. (2025) [42]	Males = 32 NCF = 133 OCF = 37 IUD = 28	WM (Bilateral change detection task)	Mid-follicular Mid-luteal	Saliva samples	No systematic differences in WM between groups nor any correlations with hormone levels Mid-follicular phase: ↓WM with increasing task difficulty
Hidalgo-Lopez and Pletzer (2021) [43]	NCF = 39	WM (n-back test) Executive function (n-back test)	Menstrual Pre-ovulatory Mid-luteal	Saliva samples	No menstrual cycle effects were observed on behavioural measures Different patterns of brain connectivity, depending on menstrual cycle phase Menstrual phase: ↓Connectivity between fronto-striatal areas and regions related to salience and cognitive effort Pre-ovulatory: Negative connectivity between fronto-striatal areas and regions related to salience and cognitive effort Mid-luteal: ↑Connectivity between fronto-striatal areas, posteromedial structures, and regions related to salience and cognitive effort
Leeners et al. (2017) [44]	First cycle: NCF = 88, Second cycle: NCF = 68	WM (block span test) Attention (Divided Attention Bimodal task) Cognitive control (Cognitive Bias test)	Pre-menstrual Mid-follicular Pre-ovulatory Mid-luteal	Blood samples Transvaginal ultrasound	Mid-follicular phase: ↓Attention in first cycle only Pre-ovulatory phase: ↑Attention in first cycle only Mid-luteal: P4 negatively correlated to WM in first cycle only No findings replicated during the second menstrual cycle
Louis et al. (2023) [45]	Met/Met ^c^ NCF = 33 Val/Val ^c^ NCF = 41	WM (n-back test)	Full menstrual cycle	Daily saliva samples	Val/Val*: when ↑E2, ↑WM (within-person) Met/Met*: when ↑E2, ↓WM (within-person) Within-person E2 levels were negatively correlated to RT
Schmalenberger et al. (2024) [46]	NCF = 19 (with suicidal ideation)	WM (n-back) Verbal fluency (Verbal fluency test) Inhibition (Stop-signal task)	Menstrual Mid-follicular Mid-luteal	Ovulation test E2 transdermal patch Placebo patch Placebo pill	(pre-)menstrual phase: ↓WM, ↓Verbal fluency in placebo condition E2 administration (regardless of additional P4) prevented decreased WM and verbal fluency performance during the (pre-)menstrual phase
Tuslyan et al. (2023a) [47]	NCF = 40	WM (dual-task n-back test)	Pre-ovulatory Mid-luteal	Self-report tracking ^a^ Backwards counting ^a^	Mid-luteal phase: ↑WM
Tuslyan et al. (2023b) [48]	NCF = 40	WM (dual-task n-back test)	Pre-ovulatory Mid-luteal	Self-report tracking ^a^ Backwards counting ^a^	Mid-luteal phase: ↑Target detection, performance remained stable No significant differences were found across 3 menstrual cycles
Bürger et al. (2024) [49]	NCF with ADHD = 10	ADHD symptoms (interviews)	Full menstrual cycle	Self-report tracking ^a^	Menstrual phase: ↑ADHD symptoms (Executive dysfunction, emotional dysregulation, inattention) Mid-luteal phase: ↑ADHD symptoms (Executive dysfunction, emotional dysregulation, inattention), ↓Effectiveness medication
De Jong et al. (2023) [50]	NCF with ADHD and co-occurring conditions = 9	ADHD symptoms and pharmacotherapy (Community case study)	Pre-menstrual	Self-report tracking ^a^	Pre-menstrual phase: ↑ADHD and depressive symptoms, ↓Effectiveness medication. Increasing the stimulant dosage during the premenstrual week improved ADHD and mood symptoms with minimal adverse events
Diekhof (2015) [51]	NCF = 28 (low trait impulsivity = 14)	Impulsivity (reward acquisition paradigm) Trait impulsivity (Barrett Impulsiveness Scale)	Menstrual Pre-ovulatory	Self-report tracking ^b^ Saliva samples	Menstrual phase: ↑Impulsive choices E2 levels correlated positively with impulsive choices during the menstrual phase, particularly in women with low trait impulsiveness
Pletzer et al. (2019) [52]	NCF = 36	Spatial navigation (landmark test) Verbal fluency (verbal fluency test)	Pre-ovulatory Mid-luteal	Saliva samples Ovulation test	No significant cognitive performance differences along the menstrual cycle. Pre-ovulatory phase: E2 boosts hippocampal activation Mid-luteal phase: P4 boosts fronto-striatal activation
Roberts et al. (2018) [53]	NCF = 32	ADHD symptoms (Current ADHD Symptoms Scale: Self-report) Trait impulsivity (Urgency, Premeditation, Perseverance, Sensation Seeking-Positive Urgency Trait Impulsivity Scale)	Full menstrual cycle	Saliva samples (daily)	Post-ovulatory phase: ↑Inattention symptoms, ↑Hyperactivity/impulsivity symptoms Mid-follicular phase: ↑Hyperactivity/impulsivity, especially with high trait impulsivity Decreased E2 + increased P4 (mid-luteal phase): ↑ADHD symptoms, especially with high trait impulsivity
Zhuang et al. (2020) [54]	NCF = 16	Impulsivity (Monetary delay discounting task) Resting-state fMRI Task-based fMRI	Pre-ovulatory Mid-luteal	Backwards counting ^a^	Pre-ovulatory phase: ↑Responsivity to short-term rewards, ↑Activity in dorsal striatum, dorsal striatum-dlPFC connectivity magnitude correlated negatively with impulsivity Mid-luteal phase: ↑DlPFC activity in rest, which was sensitive to E2 levels

^a^ Self-reported menstrual cycle phase; no hormonal verification conducted. ^b^ Self-reported menstrual cycle phase combined with hormonal verification method. ^c^ Val/Val and Met/Met are genotypes that have two copies of the valine or methionine variant of the COMT gene. Abbreviations: E2 = oestrogen, P4 = progesterone, NCF = naturally cycling females, OCF = females on oral contraceptives, RT = reaction time, WM = working memory, DA = dopamine, IUD = intrauterine device, ADHD = attention deficit hyperactivity disorder, PMDD = pre-menstrual dysphoric disorder, PMS = pre-menstrual syndrome, dlPFC = dorso-lateral prefrontal cortex.

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
