# Peer review of "Menstrual Cycle-Related Hormonal Fluctuations in ADHD: Effect on Cognitive Functioning—A Narrative Review"

_jcm, 2025, doi:10.3390/jcm15010121_

Round 1

Reviewer 1 Report

Comments and Suggestions for Authors

Review attached document

Author Response

We would like to thank you sincerely for your excellent suggestions in the review of our paper. Please review the attached document for our detailed response. 

Reviewer 2 Report

Comments and Suggestions for Authors

This review aims to examine how hormonal fluctuations across the menstrual cycle influence cognitive functioning and ADHD symptom expression in women.

The study presents a timely and effective synthesis of emerging evidence on menstrual cycle–related hormonal influences in women with ADHD with clinical relevance and theoretical insight.

In the abstract, I would suggest specifying the cognitive domains of interest and the neurobiological measurements.

The introduction establishes a functional rationale for the review. The introduction would benefit from tighter integration between neurobiological mechanisms, cognitive effects, and the study’s translational significance.

The methodology section describes the search strategy and the inclusion criteria. However, it is essential to provide additional detail enhance reproducibility and rigor. Specifically, the description of the search strategy should include information on how duplicates were removed, how study quality or risk of bias was assessed, and whether screening and data extraction were conducted independently by multiple reviewers. It is important to justify your decision to limit the search to EmBase alone without the inclusion of other major databases. Finally, the narrative synthesis approach should be briefly explained, including how findings were categorized.

The results section effectively organizes findings by cognitive domain and menstrual phase. The table is very helpful. However, its density and descriptive focus make it difficult for readers to discern overarching trends or the relative strength of the evidence. The section would benefit from more critical synthesis and integration—summarizing where findings converge or diverge, quantifying consistency across studies when possible, and distinguishing results from clinical versus non-clinical samples more clearly. Additionally, including brief interpretive comments after each subsection would enhance readability and scientific coherence.

The discussion section provides an effective synthesis of behavioral, neurochemical, and neuroimaging evidence linking menstrual cycle–related hormone fluctuations to ADHD-related cognitive processes. However, it lacks a more critical and integrative perspective. The section could also benefit from more explicit discussion of methodological heterogeneity (e.g., small sample sizes, varying phase definitions, reliance on self-reported cycle timing) and its implications for interpretation.

The limitations and future considerations section is detailed, showing critical awareness of methodological shortcomings in the literature. It would be helpful to emphasize the need for biomarker validation and explicitly propose interdisciplinary approaches combining endocrinology, neuroimaging, and computational modeling.

The clinical implications section is relevant. It effectively translates theoretical findings into practice.  Statements, such as “several trials are now underway,” should be supported by citations or clarified as preliminary. I would suggest including a paragraph on ethical considerations to strengthen the discussion’s clinical nuance.

Author Response

(The authors gave the same response as above.)

Round 2

Reviewer 1 Report

Comments and Suggestions for Authors

Dear Authors,

Thank you very much for your thorough and thoughtful response to the reviewer comments. I sincerely appreciate the substantial effort you invested in revising the manuscript and in addressing each point with clarity and precision.

Your updated version demonstrates clear methodological improvements, a more robust theoretical articulation, and greater transparency throughout. The integration of mechanistic explanations, the refinement of the Results section with quantitative summaries, the improved methodological reporting (including phase verification, screening procedures, and PRISMA), and the more nuanced discussion significantly strengthen the manuscript. I also recognize the careful attention you paid to tempering interpretations, clarifying limitations, and contextualizing clinical implications in a responsible manner.

Your detailed, point-by-point replies reflect a serious commitment to scientific rigor and greatly enhance the overall quality and coherence of the review. Thank you again for your diligent work and constructive engagement with the review process.

With best regards,

Reviewer 2 Report

Comments and Suggestions for Authors

Dear authors,

I have received your response, and I have checked the revised manuscript. My concerns have been addressed.

Kind regards